# Hyperalgesia in the Psychological Stress-Induced Fibromyalgia Model Shows Sexual Dimorphism Mediated by LPA_1_ and LPA_3_

**DOI:** 10.3390/cells14131022

**Published:** 2025-07-04

**Authors:** Hiroshi Ueda, Hiroyuki Neyama, Naoki Dozono, Junken Aoki, Jerold Chun

**Affiliations:** 1Department of Pharmacology and Therapeutic Innovation, Nagasaki University Institute of Biomedical Sciences, Nagasaki 852-8521, Japan; neyama.hiroyuki.5y@kyoto-u.ac.jp (H.N.); 44naoki7@gmail.com (N.D.); 2Department of Molecular Pharmacology, Graduate School of Pharmaceutical Sciences, Kyoto University, Kyoto 606-8501, Japan; 3Laboratory for the Study of Pain, Research Institute for Production Development, Kyoto 606-0805, Japan; 4Graduate Institute of Pharmacology, National Defense Medical Center, Nei-hu, Taipei 114201, Taiwan; 5Center for Cancer Immunotherapy and Immunobiology, Graduate School of Medicine, Kyoto University, Kyoto 606-8501, Japan; 6Department of Health Chemistry, Graduate School of Pharmaceutical Sciences, The University of Tokyo, Tokyo 113-0033, Japan; jaoki@mol.f.u-tokyo.ac.jp; 7Center for Neurological Diseases, Sanford Burnham Prebys Medical Discovery Institute, La Jolla, CA 92037, USA; jchun@sbpdiscovery.org

**Keywords:** LPA_1_, LPA_3_, fibromyalgia, empathy, psychological stress, Neurometer, splenocytes, clodronate liposome, knock-out mouse, Ki16425

## Abstract

Since the initial report indicating that LPA_1_ signaling plays a key role in initiating nerve injury-induced neuropathic pain (NeuP), subsequent studies using knockout mice and LPA_1/3_ antagonists have demonstrated that LPA_1_ and LPA_3_ signaling impact NeuP and fibromyalgia (FM) models. In the present study, we identified hyperalgesia sexual dimorphism involving LPA_1/3_ signaling in the intermittent psychological stress induced-related FM-like model called intermittent psychological stress (IPS)-induced generalized pain (IPGP) model where the hyperalgesia in IPGP mice was abolished in LPA_1_- and LPA_3_-knock-out mice. Pharmacological intervention by intraperitoneal (i.p.) treatments with the LPA_1/3_ antagonist Ki16425 consistently prevented hyperalgesia. However, intracerebroventricular treatments with Ki16425 abolished hyperalgesia in male, but not female, mice. Notably, intrathecal treatments of Ki16425 did not prevent hyperalgesia. Further studies revealed that splenocytes derived from female IPGP mice could initiate hyperalgesia via adoptive transfer in naïve mice, and this effect was abolished when donor mice were pre-treated with Ki16425 (i.p.). Thus, these studies identify male-specific LPA_1/3_-mediated mechanisms in the brain underlying IPGP, as well as distinct LPA-LPA_1/3_-mediated peripheral immune mechanisms.

## 1. Introduction

Sexual dimorphism in chronic pain has increasingly been discussed in recent years [1,2,3,4]. Fibromyalgia (FM) is overwhelmingly female-dominant and sex differences remain an important issue. However, the full picture of FM mechanisms is still not fully clear. Here, we considered the involvement of lysophosphatidic acid (LPA, a bioactive lipid mediator) receptors. Previous studies using LPA receptor 1 (LPA_1_)-KO mice have revealed that pain-related phenotypes are completely abolished in animal models of pain [5,6,7], including the following: hyperalgesia in neuropathic pain (NeuP), such as partial sciatic nerve ligation (pSNL); chemotherapeutic agent paclitaxel-induced models; type-1 and type-2 diabetes-related models; the central poststroke pain model. Furthermore, the role of LPA and LPA_1_ signaling in NeuP, has also been reported in animal models of lumbar spinal canal stenosis [8], trigeminal ganglionic compression [9], and joint neuropathy [10,11]. In the NeuP model, intense and non-selective stress produced LPA and LPA synthesis was subsequently amplified in a feed-forward manner through the activation of LPA_1_ and LPA receptor 3 (LPA_3_) on microglia [6,7]. This mechanism provides evidence that LPA plays an important role in the formation of chronic pain. Interestingly, LPA production was found to increase even 2–3 weeks after pSNL [12] and continuous administration of LPA_1/3_ antagonists completely attenuated established chronic pain. These findings support LPA_1/3_ signaling as a significant component of pain memory. Thus, the inhibitors of LPA-LPA_1_ signaling have been considered as potential targets for the treatment of NeuP [13,14].

Multiple animal models for FM have been developed for the purpose of developing diagnostics and treatments, based on symptomatology and possible pathogenesis. These include the vagotomized rat model [15]; twice repeated muscular acid injection-induced generalized pain model (AcGP) [16]; intermittent cold stress (ICS)-induced generalized pain (ICGP) model [17]; intermittent sound stress model [18]; reserpine administration-induced model [19]; intermittent psychological stress (IPS)-induced generalized pain (IPGP) model [20,21]. Recently, we reported that both IPGP and ICGP models share similar pathophysiological and pharmacotherapeutic features, which are clinically found in FM patients [21]. Among these models, we found evidence that LPA_1_ signaling is involved in the mechanisms of IPGP, ICGP, and AcGP-type FM models, by using LPA_1_-KO mice, and treatment with LPA_1_-antagonists [20,21].

Although details of molecular and cellular mechanisms of NeuP and FM models through LPA_1_ signaling remain elusive, there are reports that microglial activation through LPA-LPA_1_ signaling plays a key role in NeuP, FM, and neuroinflammation [7,22,23,24,25,26,27]. Pioneering studies have also described a sexual dimorphism in the mechanisms of NeuP. They found that microglial cells are required for NeuP in male but not female mice, whereas T-cells are likely to be involved in female mice [28,29,30,31,32,33]. In the present study, we report the sexual dimorphism that LPA_1_ and LPA_3_ mechanisms are differentially involved in the brain and periphery, including splenic immunity, using FM models.

## 2. Materials and Methods

### 2.1. Animal Experiments

Male and female C57BL/6J mice (6–12 weeks old) from Japan SLC (Shizuoka, Japan) or TEXAM Corporation (Nagasaki, Japan), and homozygous mutant mice for the LPA_1_ receptor gene (LPA_1_-KO), and LPA_3_ receptor gene (LPA_3_-KO) weighing 20–25 g were used. Male and female homozygous mutant LPA_1_- and LPA_3_-KO mice were kindly provided by Prof. Jerold Chun, co-author. LPA_1_- and LPA_3_-KO mice had been backcrossed for 10 generations with C57BL/6J, which were used for wild-type mice to match genetic background. Whilst we backcrossed, homozygous mutant litter was confirmed by genotyping, as previously reported [34,35].

They were housed at 22 ± 2 °C and 55 ± 5% relative humidity with a 12 h light/dark cycle with a standard laboratory diet and water ad libitum. All experiments were conducted in accordance with the ethical guidelines of the Kyoto University Animal Research Committees (Approval number: 19-38), Nagasaki University Animal Research Committees (Approval number: 1607201325-8), and complied with the fundamental guidelines for the proper conduct of animal experiments and related activities in academic research institutions under the jurisdiction of the Ministry of Education, Culture, Sports, Science and Technology, Japan. In addition, the present study has been performed to comply with ARRIVE guidelines [36]. For randomization, the mechanical pain thresholds of each animal were measured before experiments, and animals were assigned to the control and treatment groups, which had approximately equal pain thresholds. Investigators who oversaw the project, the experimental design, and the conduct of experiments were independent to achieve blinding. The total number of animals used in this study was 172.

### 2.2. IPGP and ICGP Model

For the IPGP model, mice were exposed to intermittent psychological stress (IPS) by using the communication box (CBX-9M, Muromachi-Kikai, Tokyo, Japan) that has nine compartments (10 cm × 10 cm) with divided transparent plastic walls. Physical electrical stress (120 times) was given to 5 mice through the grid floor by a shock generator (CSG-001, Muromachi-Kikai, Tokyo, Japan) and cycler timer (CBX-CT, Muromachi-Kikai, Tokyo, Japan). These mice were individually put in the compartment located at the center and 4 corners without cover, as previously reported [20,21]. For IPS, mice which were in the remaining 4 compartments with insulating plastic cover, were exposed to psychological stress through seeing, hearing, and smelling the foot-shocked mice. In all experimental paradigms, stress was given once per day for 5 days. The post-stress day 1 stands as P1. For ICGP, mice were exposed to intermittent cold stress (ICS) as previously described [17,21]. In this model, mice were kept in a room under the condition of 22 ± 2 °C and humidity of 60 ± 5%. During the stress period, two mice were kept in each cage (12 × 15 × 10.5 cm), and fed ad libitum with a standard laboratory diet and tap water. On the 1st day (or day 0), mice were kept in a cold room at 4 ± 2 °C from 16:30 PM. In the cold room, mice were placed on a stainless-steel floor to induce a rapid temperature change and covered with plexiglass cage. The next day, mice were removed at 10:00 AM (day 1), and kept at 24 ± 2 °C, leaving the stainless floor in the cold room. Thirty min later, mice were put back in the cold room for another 30 min. These processes were repeated until 16:30 PM, followed by the placement of mice in the cold room overnight. On the next day (day 2), the treatment of mice with alternate temperature change processes was performed again. Finally, on day 3 (or post-stress day 1/P1), mice were removed from the cold room at 10:00 AM, kept at room temperature for at least 1 h, and used for tests. When constant cold stress (CCS) was given for 3 days, mice also acquired hyperalgesia, but the hyperalgesia completely recovered within 5 days. This suggests that generalized and chronic pain induced by ICS is not simply attributed to the cold stress. There was no apparent skin damage after ICS or CCS.

### 2.3. Partial Sciatic Nerve Ligation (pSNL) Model

For the pSNL model, mice received a small skin incision exposing the right-side sciatic nerve. The dorsal half of the sciatic nerve at the upper thigh level was tightly ligated with a sterile silk suture (virgin silk thread-K, USP 9-0, Kono Seisakusho, Chiba, Japan), as previously described [12]. All procedures should be smoothly but carefully performed, minimizing systemic neuroinflammatory influences.

### 2.4. Pain Tests

In the mechanical paw pressure test, mice were acclimatized on a 6 mm × 6 mm wire-mesh grid floor for over 1 h. A mechanical stimulus was then applied to the middle of the hind paw plantar surface using a digital von Frey anesthesiometer, with a rigid tip of 0.8 mm in outer diameter (Model 2390, 90 g probe, IITC Inc., Woodland Hills, CA, USA), and a transducer indicator (Model 1601; IITC Inc., Woodland Hills, CA, USA). The mechanical paw withdrawal threshold was determined from the average of three trials of paw pressure. A cut-off pressure of 20 g was set to avoid tissue damage. In the electrical stimulation-induced paw withdrawal (EPW) test, on the other hand, electrodes of Neurometer^®^ Current Perception Threshold/C (CPT/C; Neurotron Inc., Baltimore, MD, USA) were attached to the planter and insteps of the hind paw, and transcutaneous nerve stimuli with sine-wave pulses of 2000 Hz or 250 Hz were applied. The minimum intensity at which each mouse withdrew its paw (cutoff time: 3 s) was evaluated by nociceptive current threshold (μA), as described previously [21]. The in vivo patch-clamp recording studies using acutely isolated rat spinal cord slices characterized that transcutaneous stimuli at 2000 Hz and 250 Hz caused excitatory synaptic responses in the substantia gelatinosa neurons, which correspond to Aβ- and Aδ-fiber stimulation, respectively [37]. This method has an advantage that no acclimatization time is necessary since the experimenter holds the mouse.

### 2.5. Clodronate Liposome Treatments

To deplete peripheral macrophages, dichloromethylene diphosphonate (clodronate) encapsulated liposomes (0.05 mg/10 µL; HYGIEIA BIOSCIENCE, Osaka, Japan) or PBS vehicle control liposomes were administrated by i.p. injection at a volume of 200 μL/mouse, as previously described [38].

### 2.6. Isolation and Adoptive Transfer of Splenocytes

To investigate peripheral immunity contribution to hyperalgesia in female mice with IPGP, the adoptive transfer of splenocytes from a donor mouse into a naïve mouse was attempted on the analogy of previous studies [27,39]. The spleen was isolated from IPS-treated female mice at P19 and placed in 3 mL of ice-cold RPMI 1640 medium (Gibco, Grand Island, NY, USA) containing 2% fetal bovine serum (FBS), followed by mincing using the plunger of a 1 mL injection syringe. The cell suspension was filtrated through a 70 μm mesh-cell strainer, washed with 5 mL of ice-cold PBS containing 2% FBS, and incubated in 3 mL of red blood cell lysis buffer (Abcam, San Diego, CA, USA) for 3 min at room temperature. The reaction was stopped by adding 5 mL of ice-cold PBS containing 2% FBS, washed, and resuspended in 3 mL of the same buffer. An aliquot of the suspension (200 μL, 1 × 10^6^ cells) was then injected into the right retro-orbital sinus (retro-orbital injection) of the naïve male mouse under isoflurane (4%) anesthesia [40,41].

### 2.7. Drug Treatments

LPA_1/3_ antagonist Ki16425 was dissolved in saline for the intraperitoneal (i.p.) injection (100 μL/10 g of body weight) or in the aCSF (125 mM NaCl, 3.8 mM KCl, 1.2 mM KH_2_PO_4_, 26 mM NaHCO_3_, 10 mM glucose, pH 7.4) for the intracerebroventricular (i.c.v., 5 μL) or intrathecal (i.t., 5 μL) injection. The i.c.v. injection was performed using the method from Haley and McCormick [42]. The i.t. injection was given into the space between spinal L5 and L6 segments according to the method described by Hylden and Wilcox [43].

### 2.8. Statistical Analysis

All data are shown as mean ± S.E.M. Statistical value was calculated by GraphPad Prism 8 (GraphPad Software Inc., La Jolla, CA, USA) with *p* value set at under 0.05. Sample size was decided by our previous reports [7]. Statistical differences between the group were analyzed using two-way repeated measures ANOVA followed by Bonferroni’s or Tukey’s multiple comparisons test. Individual statistics were presented in figure legends.

## 3. Results

### 3.1. Involvement of LPA_3_ Signaling in IPGP and ICGP Models in Male Mice


It has already been reported that LPA_1_ plays a key role in IPGP and ICGP models using male LPA_1_-KO mice. As shown in Figure 1A,B, male wild-type (WT) mice exhibited stable mechanical hyperalgesia from P1 to P22, after IPS induction from D1 to D5. Mechanical hyperalgesia was completely absent in male LPA_3_-KO mice with IPS treatment and LPA_3_-KO mice without IPS treatment. In addition to the equivalent pathophysiological and pharmacotherapeutic features between IPGP and ICGP models, a similar anti-hyperalgesia role of LPA_1_ [20] and LPA_3_ was found in the ICGP model (Figure 1C,D). Mechanical hyperalgesia was observed in male WT mice from P1 to P19 after ICS treatment. Male LPA_3_-KO mice, in contrast, did not show mechanical hyperalgesia, either with ICS treatment or without ICS treatment.

Next, male mice were administered the LPAR_1/3_ antagonist Ki16425 30 mg/kg (i.p.), once daily from P5 to P18 (Figure 1E). The IPGP-induced mechanical hyperalgesia reversed from P12 to P19 (Figure 1F). To test the noxious paw-withdrawal response (hypersensitivity), mice received a 2000 Hz current for non-noxious Aβ fibers, and 250 Hz current for noxious Aδ fibers generated by a Neurometer^®^ [44]. The IPS-induced hypersensitivity at P1 and P5 was significantly reversed at P12 and P19 after repeated Ki16425 treatment (Figure 1G,H).

### 3.2. Involvement of LPA_1_ and LPA_3_ Signaling in the IPGP Model in Female Mice


IPS-induced pain remained stable until at least P19 in female WT mice (Figure 2B–D). In LPA_1_-KO mice with IPS, mechanical hyperalgesia was not observed at P1 and P5. In LPA_3_-KO mice, mechanical hyperalgesia was not observed at P1, but a reduction in paw withdrawal threshold occurred at P5 suggesting partial hyperalgesia suppression may occur, although without significance. When Ki16425 was administered continuously in WT mice from P5 to P18 (Figure 2A), mechanical hyperalgesia was significantly blocked at P12 and P19 (Figure 2B). Similar observations were also observed with Neurometer^®^ stimulation experiments. LPA_1_-KO mice did not develop hypersensitivity at P1 and P5, evaluated by 2000 Hz and 250 Hz stimulation (Figure 2C,D). LPA_3_-KO mice did not show hypersensitivity at P1 with 2000 Hz stimulation but did display hypersensitivity at P5. At 250 Hz the anti-hypersensitivity effect was partial but significant at P1 and P5. Thus, the blocking of mechanical hyperalgesia and Neurometer hypersensitivity in LPA_3_-KO mice appears to be less evident than in LPA_1_-KO mice. Therefore, LPA_3_ may be more important in the early or developing stage of abnormal pain, but less so in the maintenance stage. When Ki16425 was i.p. administered, hypersensitivity to the stimulation with 2000 Hz and 250 Hz at P12 and P19 significantly decreased. The reversal of hypersensitivity was more evident at the later time point, P19.

### 3.3. Sex Differences in the Effect of Intracerebroventricular Administration of Ki16425


Next, Ki16425 at 10 nmol was administered intracerebroventricularly (i.c.v.) to male mice once daily from P5 to P11 (Figure 3A). At P12, IPS-induced mechanical hyperalgesia, and Neurometer hypersensitivity at 2000 Hz and 250 Hz were significantly blocked (Figure 3B–D). Surprisingly, Ki16425 (i.c.v.) administration to female mice showed no anti-hyperalgesia or anti-hypersensitivity effect (Figure 3E–G).

### 3.4. Lack of Blocking Effects on Hyperalgesia and Hypersensitivity by Intrathecal Administration of Ki16425


Ki16425 at 10 nmol was then intrathecally (i.t.) administered to male and female mice daily from P5 to P11 (Figure 4A). No effect in response to treatment was observed in mechanical hyperalgesia, or Neurometer hypersensitivity at 2000 Hz and 250 Hz, up until P12 (Figure 4B–G).

### 3.5. Lack of Blocking Effect of Clodronate Liposome on Female IPGP Model


It is known that i.p. treatment with clodronate liposome depletes peripheral macrophage to block neuropathic pain [38]. To evaluate effects of clodronate liposome treatment in the present study, liposomes containing 25 mg/kg of clodronate were given 30 min before and 6 days after the pSNL (Figure 5A,B). The mechanical hyperalgesia at the ipsilateral paw of male mice at P1 and P7 was abolished by clodronate liposome treatment (Figure 5C). The hypersensitivity evaluated by 2000 Hz and 250 Hz stimulation was also significantly reversed by clodronate liposome treatment (Figure 5D,E).

In the IPGP model in female mice, clodronate liposomes were given 30 min before the first day of IPS and after day 5 of IPS (Figure 5F,G). The mechanical hyperalgesia or Neurometer hypersensitivity evaluated by 2000 Hz and 250 Hz stimulation at P1 and P5 after IPS was not affected by clodronate liposomes (Figure 5H–J).

### 3.6. Blocking Effect of Ki16425 on the Induction of Pain Hypersensitivity by Splenocytes Derived from Female IPGP Model Mice


Next, splenocytes were prepared at P19 after the IPS in female mice and were then administered (i.v.) to naïve mice (Figure 6A,B). This induced significant mechanical hyperalgesia or electrical hypersensitivity evaluated by 2000 Hz and 250 Hz stimulation at P1 (Figure 6C–E). The hyperalgesia and hypersensitivity were significantly reversed by repeated Ki16425-treatments (i.p.), from P5 to P18 in donor mice (Figure 6C–E).

## 4. Discussion

Previous reports have shown that thermal hyperalgesia in IPGP and ICGP models are completely abolished in male LPA_1_-KO mice [20]. The present study evaluated the sex-dependent involvements of LPA_1_ and LPA_3_ in the IPGP model. Mechanical paw withdrawal threshold was tested using an electronic digital von Frey anesthesiometer and the electrical stimulation-induced paw withdrawal test (EPW) was performed using a Neurometer, as briefly summarized in Table 1. In the IPGP model, male LPA_1_- or LPA_3_-KO mice did not experience mechanical hyperalgesia and EPW hypersensitivity. Similar effects were observed in male LPA_1_- or LPA_3_-KO mice after systemic administration (i.p.) of Ki16425 (an LPA_1/3_ antagonist). In female LPA_1_-KO mice, mechanical hyperalgesia and EPW hypersensitivity were also absent. Whilst in LPA_3_-KO mice, there was no, or a very weak blocking effect, even when treated with Ki16425 (i.p.). Sexual dimorphism of LPA_1/3_ receptor signaling was more evident when Ki16425 was administered into the brain (i.c.v.). Repeated i.c.v. treatments with Ki16425 completely reversed the mechanical hyperalgesia and EPW hypersensitivity in male mice, but no reversal was observed in female mice. As Ki16425 (i.t.) had no effect on male and female mice, brain LPA_1_ and LPA_3_ signaling likely play key roles in the mechanisms underlying IPGP in male mice. Female LPA_3_ mice, in contrast, show no, or very limited contribution of LPA_3_ signaling. There are reports that LPA_1_ and LPA_3_ signaling are associated with activating microglia, namely for the self-amplification of LPA and production of brain-derived neurotrophic factor in the NeuP mechanisms [7,45]. All these findings are reminiscent of previous reports that NeuP is associated with microglial activation in male, but not female mice [32,33].

There have been many reports on the role of macrophages in NeuP models [46]. One experimental evidence for this can be seen in a study showing the disappearance of NeuP by using clodronate liposomes, which phagocytose activated peripheral macrophages and cause cell death via metabolic toxin clodronate [38]. In the present study, we verified the validity of this technique using the pSNL model. As reported before [47], the present study confirmed that clodronate liposome treatment abolished pSNL-induced hyperalgesia and hypersensitivity. However, clodronate liposome treatment had no effect in the IPGP model at P1 and P5. Therefore, it is evident that peripheral macrophages are not involved in IPGP mechanisms.

Accumulating studies have reported that the peripheral immune system is involved in the sexual dimorphism of NeuP mechanisms. These suggest brain microglia are involved in male, but not female mice. In female mice, the peripheral immune system, but not central, likely plays a key role in the NeuP mechanism [30,33]. To further investigate a possible involvement of the peripheral immune system, we explored the involvement of splenocytes. This was based on the analogy of our previous study, in which splenocytes derived from donor mice with chronic muscular pain induced by twice repeated acid injections, caused hyperalgesia in naïve mice [27]. The present study confirmed this speculation by finding that systemic (retro-orbital) injection of splenocytes from donor mice in the IPGP model, into naïve mice, caused significant hyperalgesia and hypersensitivity. As systemic Ki16425-treatment of female donor mice reversed the abnormal pain activity of splenocytes, peripheral LPA_1/3_ mechanisms are likely involved in this reversal. However, it remains unclear whether LPA_1/3_ signaling in splenocytes is responsible for such mechanisms. For this discussion, it is necessary to perform several key experiments in female mice treated with IPS, such as whether LPA_1/3_ or biosynthesis enzyme autotaxin are upregulated in splenic T-cells or B-cells. Current reports suggest that plasma cytokine levels (possibly from T-cells) in FM patients may be associated with FM development [48,49,50]. Furthermore, pain-related immunoglobulin G (IgGs) may be derived plasma cells differentiated from activated B-cells [51]. A recent study also revealed that IgGs from FM patients were accumulated in satellite glial cells, possibly leading to DRG neuron stimulation [52,53,54].

It remains unclear what mechanisms are involved in male brain-specific LPA_1_ and LPA_3_ mechanisms in the IPGP model. Future studies on male-specific microglial roles, as seen in the case of AcGP-type FM model [27], may be noteworthy. Indeed, recent studies demonstrated the male-specific expression of microglia-derived molecules, which are associated with sexually dimorphic NeuP [55,56,57]. Thus, it should be interesting to investigate whether LPA_1_ or LPA_3_ signaling is associated with the actions on these specific molecules.

## 5. Conclusions

In conclusion, accumulating studies demonstrated that LPA_1/3_ signaling plays key roles in the mechanisms underlying several NeuP and FM models [7]. The present study firstly demonstrated that there is sexual dimorphism in LPA_1/3_ signaling in the IPGP model. Upon i.c.v. treatment with an LPA_1/3_-antagonist, hyperalgesia and hypersensitivity in male, but not female mice, were abolished, although systemic treatment reversed the abnormal pain in both male and female mice. Furthermore, studies revealed that peripheral macrophages were not involved, whilst splenocytes derived from female IPGP model mice caused hyperalgesia in naïve mice. Thus, these findings advocate the necessity to identify the male-specific LPA_1/3_-mediated mechanisms in the brain underlying IPGP, as well as molecular-based peripheral immune mechanisms, which may contribute to the translational potential of the research, such as diagnostics and treatments by targeting involved molecules.

## Figures and Tables

**Figure 1 cells-14-01022-f001:**
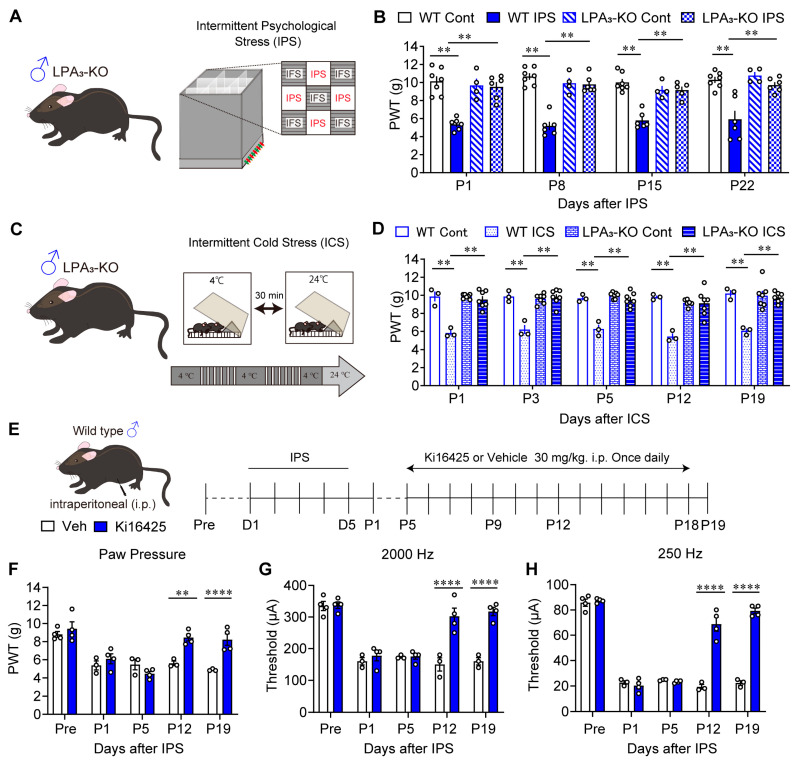
Reversal of male IPGP and ICGP by genetic deletion of LPA_3_ or repeated i.p. treatments with LPA_1/3_ antagonist. (**A**) Illustrative diagram of intermittent psychological stress (IPS)-induced generalized pain (IPGP) model. (**B**) Reversal of IPS-induced mechanical hyperalgesia by genetic deficiency of LPA_3_ in male mice. Mechanical hyperalgesia at P1–P22 was observed in the paw withdrawal test. (**C**) Illustrative diagram of intermittent cold stress (ICS)-induced generalized pain (ICGP) model. (**D**) Reversal of ICS-induced mechanical hyperalgesia (P1–P19) by genetic deficiency of LPA_3_ in male mice. (**E**) Schedule of IPS-treatments and systemic (i.p.)-treatments with Ki16425. (**F**–**H**) Time-dependent change in threshold in the IPGP model treated with vehicle or Ki16425 in the paw pressure test (**F**), EPW 2000 Hz (**G**), and EPW 250 Hz (**H**). (**B**) ** *p* < 0.01, two-way ANOVA followed by Tukey’s multiple comparisons test (WT-Cont, *n* = 6; WT-IPS, *n* = 6; LPA_3_-KO-Cont, *n* = 4; LPA_3_-KO-IPS, *n* = 5). (**D**) ** *p* < 0.01, two-way ANOVA, followed by Tukey’s multiple comparisons test (WT-Cont, *n* = 3; WT-ICS, *n* = 3; LPA_3_-KO-Cont, *n* = 7; LPA_3_-KO-ICS, *n* = 7). (**F**–**H**) ** *p* < 0.01, **** *p* < 0.0001, two-way ANOVA followed by Bonferroni’s multiple comparisons test (Veh, *n* = 3–4; Ki16425, *n* = 4).

**Figure 2 cells-14-01022-f002:**
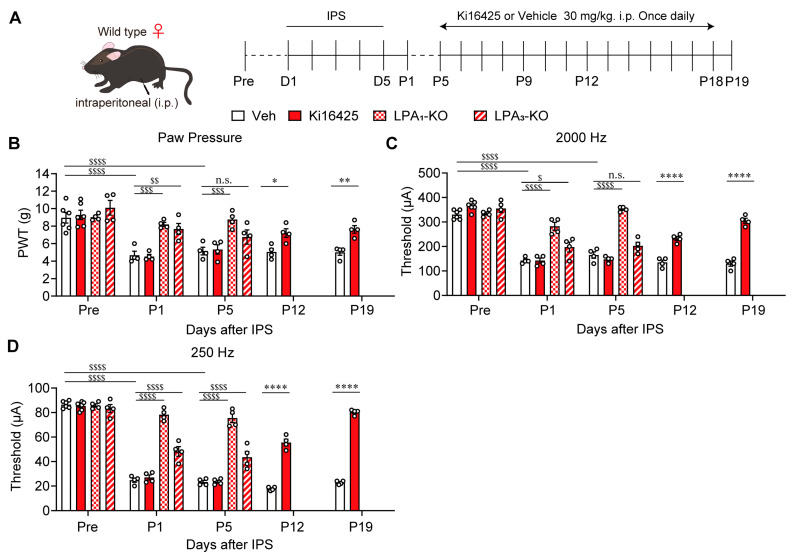
Reversal of female IPGP by genetic deletion of LPA_1/3_ or repeated i.p. treatments with Ki16425. (**A**) Schedule of IPS-treatments and systemic (i.p.)-treatments with Ki16425. (**B**–**D**) Time-dependent reversal of IPS-induced mechanical hyperalgesia (**B**) and hypersensitivity in EPW tests with 2000 Hz (**C**) or 250 Hz (**D**) by genetic deficiency of LPA_1_ or LPA_3_, or by Ki16425 (i.p.) in female mice. (**B**–**D**) * *p* < 0.05, ** *p* < 0.01, **** *p* < 0.0001, Veh vs. Ki16425 (0–19), and two-way ANOVA followed by Bonferroni’s multiple comparisons test (Veh, *n* = 4–6; Ki16425, *n* = 4–6). ^$^ *p* < 0.05, ^$$^ *p* < 0.01, ^$$$^ *p* < 0.001, ^$$$$^ *p* < 0.0001, n.s.; not significant, Veh vs. LPA_1_-KO or LPA_3_-KO, nd two-way ANOVA followed by Tukey’s multiple comparisons test (0–5; Veh, *n* = 4–6; LPA_1_-KO, *n* = 4; LPA_3_-KO, *n* = 4).

**Figure 3 cells-14-01022-f003:**
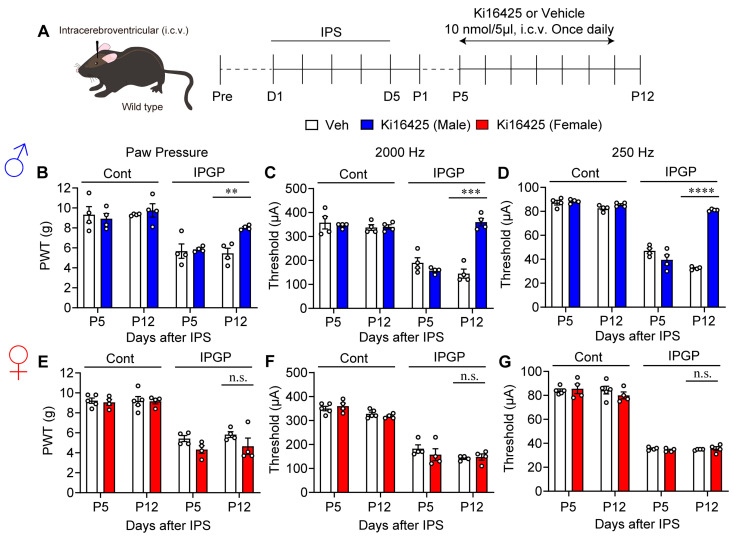
Male-specific reversal of hyperalgesia and hypersensitivity by repeated i.c.v. treatments with Ki16425. (**A**) Schedule of IPS-treatments and repeated i.c.v. treatments with Ki16425. (**B**–**D**) Reversal of IPS-induced mechanical hyperalgesia (**B**) and hypersensitivity in EPW tests with 2000 Hz (**C**) or 250 Hz (**D**) by Ki16425 (i.c.v.)-treatments in male mice. (**E**–**G**) Lack of reversal of IPS-induced mechanical hyperalgesia (**E**) and hypersensitivity in EPW tests with 2000 Hz (**F**) or 250 Hz (**G**) by Ki16425 (i.c.v.)-treatments in female mice. (**B**–**G**) ** *p* < 0.01, *** *p* < 0.001, **** *p* < 0.0001, n.s.; not significant, and two-way ANOVA followed by Bonferroni’s multiple comparisons test (male Veh, *n* = 4; female Veh, *n* = 5; Ki16425, *n* = 4).

**Figure 4 cells-14-01022-f004:**
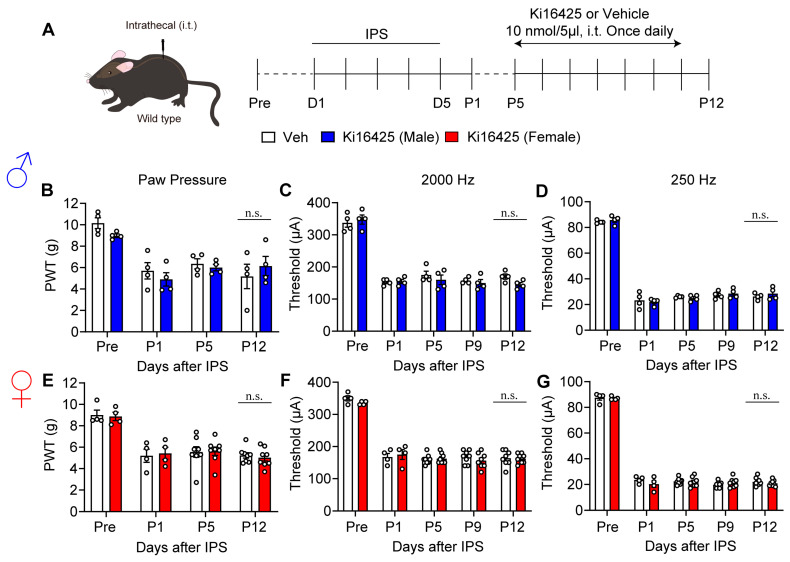
Lack of reversal of hyperalgesia and hypersensitivity by i.t. treatments with Ki16425. (**A**) Schedule of IPS-treatments and repeated i.t. treatments with Ki16425. (**B**–**G**) Lack of reversal of IPS-induced mechanical hyperalgesia (**B**,**E**) and hypersensitivity in EPW tests with 2000 Hz (**C**,**F**) or 250 Hz (**D**,**G**) by Ki16425 (i.t.)-treatments in male (**B**–**D**) and female mice (**E**–**G**). (**B**–**G**) n.s.; not significant and two-way ANOVA followed by Bonferroni’s multiple comparisons test (male Veh, *n* = 4; female Veh, *n* = 4–8; Ki16425, *n* = 4–8).

**Figure 5 cells-14-01022-f005:**
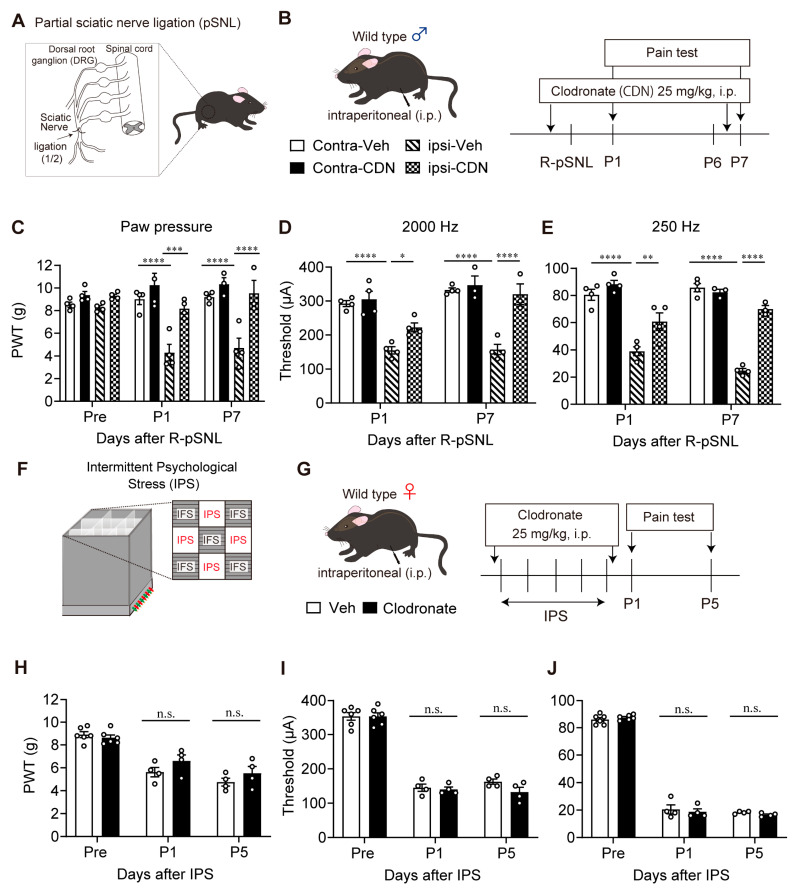
Lack of reversal of IPS-induced hyperalgesia and hypersensitivity by clodronate liposome in the IPGP model. (**A**) Illustrative diagram of partial sciatic nerve injury (pSNL) model. (**B**) Schedule of clodronate treatments and pain tests in the partial sciatic nerve injury (pSNL) model in male mice. (**C**–**E**) Reversal of pSNL-induced mechanical hyperalgesia (**C**) or hypersensitivity in the EPW 2000 Hz (**D**) or 250 Hz (**E**) tests by clodronate liposome treatments. (**F**) Illustrative diagram of IPGP model. (**G**) Schedule of clodronate treatments and pain tests in the IPGP model in female mice. (**H**–**J**) Lack of reversal of the IPS-induced hyperalgesia (**H**) or hypersensitivity (**I**,**J**) by clodronate liposome treatments. (**C**–**E**) * *p* < 0.05, ** *p* < 0.01, *** *p* < 0.001, **** *p* < 0.0001, and two-way ANOVA followed by Tukey’s multiple comparisons test (contra-Veh, *n* = 4; ipsi-Veh, *n* = 4; contra-Clodronate, *n* = 3–4; ipsi-Clodronate, *n* = 3–4). (**H**–**J**) n.s.; not significant and two-way ANOVA followed by Bonferroni’s multiple comparisons test (Veh, *n* = 4–6; Clodronate, *n* = 4–6).

**Figure 6 cells-14-01022-f006:**
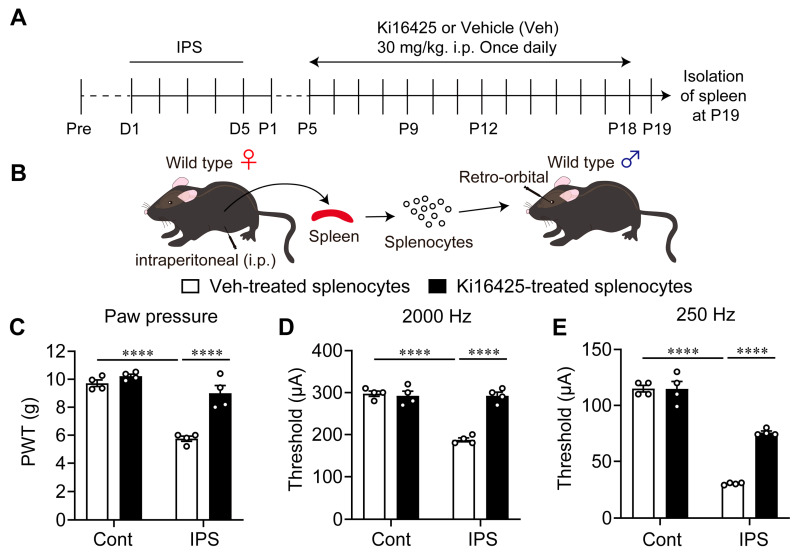
Abnormal pain by splenocytes derived from IPS-treated mice. (**A**) Schedule of IPS treatments and repeated i.p. treatments with Ki16425 in female donor mice. (**B**) Experimental procedure for splenocytes injection. (**C**–**E**) Ki16425-reversible hyperalgesia (**B**) and hypersensitivity (**C**,**D**) in naïve female mice by adoptive transfer of splenocytes from IPS-treated donor female mice. Ki16425 (i.p.) was treated in donor mice, as described in the legend of Figure 3. (**C**–**D**) **** *p* < 0.0001 and two-way ANOVA followed by Tukey’s multiple comparisons test (*n* = 4).

**Table 1 cells-14-01022-t001:** Summary of LPA_1_ and LPA_3_-mediated hyperalgesia and hypersensitivity.

IPS Model	Sex	Hyperalgesia (Paw Pressure)	Hypersensitivity (2000 Hz)	Hypersensitivity (250 Hz)
LPA_1_-KO	Male	− ^(1)^	No data	No data
	Female	−	−	−
LPA_3_-KO	Male	− ^(2)^	No data	No data
	Female	−	−	−
WT Ki 16425, i.p.	Male	−	−	−
	Female	−	−	−
WT Ki 1642, i.c.v.	Male	−	−	−
	Female	+	+	+
WT Ki 16425, i.t.	Male	+	+	+
	Female	+	+	+

^(1)^: Data have been reported elsewhere [20]. ^(2)^: The blockade of hyperalgesia was also observed in the ICS model.

## Data Availability

Data are available upon request to the authors.

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
