# Peer review of "Hyperalgesia in the Psychological Stress-Induced Fibromyalgia Model Shows Sexual Dimorphism Mediated by LPA1 and LPA3"

_cells, 2025, doi:10.3390/cells14131022_

Round 1
Reviewer 1 Report
Comments and Suggestions for Authors
In an article by Ueda et al, an attempt was made to determine sexual dimorphism in LPA1 and LPA3-mediated chronic hyperalgesia using a model of empathy-related fibromyalgia. Given that fibromyalgia is clinically manifested mainly in women, the importance of this work is undeniable. In addition, the team is a pioneer in the study of LPA signaling during the occurrence of chronic pain in various models, including fibromyalgia. Overall, the authors have empirically proved the existence of sex-related LPA1-3 dimorphism, which likely contributes to the prevalence of fibromyalgia in women. Although the manuscript is of great clinical importance, some shortcomings limit the perception and understanding of the presented results.
- The manuscript is very difficult to write. English must be adapted for better understanding.
- The term "empathy-related fibromyalgia" is controversial in the context of the fact that mice experienced stress due to vision, hearing, and smell, i.e. physical factors. The development of hyperalgesia is not a sign of empathy. Empathy in animals is often linked to social bonds, altruistic behaviors, and cognitive abilities.
Methods:
- How is intermittent cold stress related to fibromyalgia? Two nights spent on a stainless-steel floor at 2°C can cause skin damage underlying hyperalgesia. Were the mice tested before the behavior experiments?
Results:
- 1 The links to the figures (1A-G) are mixed up.
- What additional information was obtained when using the intraperitoneal LPA1-3 antagonist Ki16425 compared to LPA3-KO mice, given that this antagonist is non-selective and more potent to LPA1? Why was the 30 mg/kg dose chosen?
- How would the authors interpret an increase in the EPW 2000 Hz stimulation in LPA3-KO mice? How is this related to hyperalgesia?
- 2 The representation of significance in Figure 2B-D is confusing.
- 5 This part of the results is very confusing.
- Why has the effectiveness of liposomes with clodronate been tested only on male mice with pSNL, but not on male mice with IPS? The lack of such experiments limited the interpretation of the results regarding sex-related dimorphism.
- 6 Additional information is needed to clarify the method of splenocyte administration.
- The role of splenocytes was considered as a possible mechanism, although it needs to be more thoroughly substantiated.
- There is no correspondence between the legend and Figure 6A about the donor and recipient of the splenocytes.
The discussion should be adapted to the new results.

Author Response
#Reviewer 1
Thank you very much for valuable comments. According to the reviewer’s comments, we revised the manuscript.
Comments and Suggestions for Authors
In an article by Ueda et al, an attempt was made to determine sexual dimorphism in LPA1 and LPA3-mediated chronic hyperalgesia using a model of empathy-related fibromyalgia. Given that fibromyalgia is clinically manifested mainly in women, the importance of this work is undeniable. In addition, the team is a pioneer in the study of LPA signaling during the occurrence of chronic pain in various models, including fibromyalgia. Overall, the authors have empirically proved the existence of sex-related LPA1-3 dimorphism, which likely contributes to the prevalence of fibromyalgia in women. Although the manuscript is of great clinical importance, some shortcomings limit the perception and understanding of the presented results.
- The manuscript is very difficult to write/read. English must be adapted for better understanding.
>>As suggested, we asked the British native and pain scientist (Dr. Sant-Cassia, Myles) for the English edition. Reviewer 2 also kindly suggested the corrections at some parts of the manuscript.
- The term "empathy-related fibromyalgia" is controversial in the context of the fact that mice experienced stress due to vision, hearing, and smell, i.e. physical factors. The development of hyperalgesia is not a sign of empathy. Empathy in animals is often linked to social bonds, altruistic behaviors, and cognitive abilities.
>>According to the reviewer’s comment, we replaced the term ‘empathy-related’ with ‘intermittent psychological stress (IPS)-induced’. (Title, lines 22-23)
- How is intermittent cold stress related to fibromyalgia? Two nights spent on a stainless-steel floor at 2°C can cause skin damage underlying hyperalgesia. Were the mice tested before the behavior experiments?
>>In the revised manuscript, we described the validity of ICS to induce a generalized pain (ICGP). The ICGP mimics pathophysiological and pharmacotherapeutic features in FM patients, as in the case with IPS-induced generalized pain (IPGP). Details have been reported in Biomedicines [1]. In this report, we described that constant cold stress (CCS) at 4 ± 2℃ for three days also causes hyperalgesia, but the hyperalgesia completely recovers within 5 days. These findings suggest that the generalized pain induced by ICS is not simply attributed to the cold stress. There was no apparent skin damage after the ICS or CCS.
(line 124-127)
- 1 The links to the figures (1A-G) are mixed up. What additional information was obtained when using the intraperitoneal LPA1-3 antagonist Ki16425 compared to LPA3-KO mice, given that this antagonist is non-selective and more potent to LPA1? Why was the 30 mg/kg dose chosen?
>>In the present study, we successfully enabled to show the therapeutic potentials or maintenance roles of chronic pain by LPA1/3 antagonist, not merely the pathophysiological roles of LPA1/3 in the IPGP model, which are obtained using knock-out mice. We have used Ki16425 at 30 mg/kg i.p. to see effects on peripheral organs, brain and spinal cord in several reports in neuropathic pain models [2,3]. The choice of this dose would be convenient to compare the potency of Ki16425 between neuropathic pain and fibromyalgia models.
- How would the authors interpret an increase in the EPW 2000 Hz stimulation in LPA3-KO mice? How is this related to hyperalgesia?
>>In Fig. 1, mechanical hyperalgesia in the IPGP model was completely abolished in male LPA3-KO mice, while mechanical hyperalgesia and hypersensitivity to electrical (2000 Hz or Aβ) stimulation was only partially blocked in female LPA3-KO mice. As the roles of brain LPA1/3 receptor signaling were not evidenced in female IPGP model mice, peripheral LPA3 signaling may play weak roles in the hypersensitivity to 2000 Hz stimulation. In addition, LPA3 may play roles in the early or development stage, but not in the maintenance stage of abnormal pain (line 237-238).
- The representation of significance in Figure 2B-D is confusing.
>>According to the reviewer’s comment, we revised the significance, as follows.
(B-D) *P<0.05, **P<0.01, ****P<0.0001, Veh vs. Ki16425 (0-19), two-way ANOVA followed by Bonferroni’s multiple comparisons test (Veh, n=4-6; Ki16425, n=4-6). $P<0.05, $$$P<0.001, $$$$P<0.0001, n.s.; not significant, Veh vs. LPA1-KO or LPA3-KO, two-way ANOVA followed by Tukey’s multiple comparisons test (0-5; Veh, n=4-6; LPA1-KO, n=4; LPA3-KO, n=4). (line 247-248)
- This part of the results is very confusing. Why has the effectiveness of liposomes with clodronate been tested only on male mice with pSNL, but not on male mice with IPS? The lack of such experiments limited the interpretation of the results regarding sex-related dimorphism.
>>The study using liposomes in the pSNL model was not intended to see the sex-difference. We performed this study to confirm the effectiveness of clodronate liposomes to deplete the actions of peripheral macrophages, which have been often reported by other investigators. As the hyperalgesia and hypersensitivity were not affected by this treatment in the IPGP model, we just showed the validity of this treatment. (line 345-348)
- Additional information is needed to clarify the method of splenocyte administration.
>>We added the necessary information. An aliquot of the suspension (200 μL, 1 × 106 cells) was injected into the right retro-orbital sinus (retro-orbital injection) of the naïve male mouse under the isoflurane (4%) anesthesia [4].[https://www.queensu.ca/animals-in-science/policies-procedures/sop/mice/7-11] (line 172-174).
- The role of splenocytes was considered as a possible mechanism, although it needs to be more thoroughly substantiated.
>>The present study did not describe which molecules derived from splenocytes or which cell types of splenocytes are involved in causing the hyperalgesia to naïve mice. Regarding the former possibility, there are reports that IgGs from FM patients caused hyperalgesia in mice, possibly through an activation of satellite glial cells and subsequent stimulation of DRG neurons. However, details remain whether or how splenic cells are involved in the pain-producing IgGs. Regarding the latter possibility, on the other hand, plasma levels of cytokines are upregulated in FM patients. Regarding this issue, we added following sentences.
For this discussion, it is necessary to perform several key experiments whether LPA1/3 or biosynthesis enzyme autotaxin are upregulated in splenic T-cells or B-cells in female mice treated with IPS, since there are reports that plasma cytokine levels, possibly from T-cells in FM patients seem to be associated to the FM development [5-7], and pain-related IgGs may be derived plasma cells differentiated from activated B cells [8]. Recent study revealed that IgGs from FM patients are accumulated in satellite glial cells, possibly leading to a stimulation of DRG neurons [9-11]. (line 364-370)
- There is no correspondence between the legend and Figure 6A about the donor and recipient of the splenocytes.
>>We revised the Fig. 6 and related legends. (line 310)
- The discussion should be adapted to the new results.
>>According to the comment, we added some explanations/discussions of findings to Results section, to avoid repetitive statements in the Discussion section. Instead, we revised the first paragraph of Discussion section to give an overview of the new results about sexual dimorphism and included the possible relationship of peripheral immunity citing previous studies.
References
- Ueda, H.; Neyama, H. Fibromyalgia Animal Models Using Intermittent Cold and Psychological Stress. Biomedicines 2023, 12, doi:10.3390/biomedicines12010056.
- Ma, L.; Matsumoto, M.; Xie, W.; Inoue, M.; Ueda, H. Evidence for lysophosphatidic acid 1 receptor signaling in the early phase of neuropathic pain mechanisms in experiments using Ki-16425, a lysophosphatidic acid 1 receptor antagonist. J Neurochem 2009, 109, 603-610, doi:10.1111/j.1471-4159.2009.05987.x.
- Ueda, H.; Neyama, H.; Sasaki, K.; Miyama, C.; Iwamoto, R. Lysophosphatidic acid LPA(1) and LPA(3) receptors play roles in the maintenance of late tissue plasminogen activator-induced central poststroke pain in mice. Neurobiol Pain 2019, 5, 100020, doi:10.1016/j.ynpai.2018.07.001.
- Yardeni, T.; Eckhaus, M.; Morris, H.D.; Huizing, M.; Hoogstraten-Miller, S. Retro-orbital injections in mice. Lab Anim (NY) 2011, 40, 155-160, doi:10.1038/laban0511-155.
- Ellergezen, P.; Alp, A.; Cavun, S.; Celebi, M.; Macunluoglu, A.C. Pregabalin inhibits proinflammatory cytokine release in patients with fibromyalgia syndrome. Arch Rheumatol 2023, 38, 307-314, doi:10.46497/ArchRheumatol.2023.9517.
- Gerdle, B.; Dahlqvist Leinhard, O.; Lund, E.; Lundberg, P.; Forsgren, M.F.; Ghafouri, B. Pain and the biochemistry of fibromyalgia: patterns of peripheral cytokines and chemokines contribute to the differentiation between fibromyalgia and controls and are associated with pain, fat infiltration and content. Front Pain Res (Lausanne) 2024, 5, 1288024, doi:10.3389/fpain.2024.1288024.
- Peck, M.M.; Maram, R.; Mohamed, A.; Ochoa Crespo, D.; Kaur, G.; Ashraf, I.; Malik, B.H. The Influence of Pro-inflammatory Cytokines and Genetic Variants in the Development of Fibromyalgia: A Traditional Review. Cureus 2020, 12, e10276, doi:10.7759/cureus.10276.
- Goebel, A.; Krock, E.; Gentry, C.; Israel, M.R.; Jurczak, A.; Urbina, C.M.; Sandor, K.; Vastani, N.; Maurer, M.; Cuhadar, U.; et al. Passive transfer of fibromyalgia symptoms from patients to mice. J Clin Invest 2021, 131, doi:10.1172/JCI144201.
- Fanton, S.; Menezes, J.; Krock, E.; Sandstrom, A.; Tour, J.; Sandor, K.; Jurczak, A.; Hunt, M.; Baharpoor, A.; Kadetoff, D.; et al. Anti-satellite glia cell IgG antibodies in fibromyalgia patients are related to symptom severity and to metabolite concentrations in thalamus and rostral anterior cingulate cortex. Brain Behav Immun 2023, 114, 371-382, doi:10.1016/j.bbi.2023.09.003.
- Jakobsson, J.E.; Menezes, J.; Krock, E.; Hunt, M.A.; Carlsson, H.; Vaivade, A.; Emami Khoonsari, P.; Agalave, N.M.; Sandstrom, A.; Kadetoff, D.; et al. Fibromyalgia patients have altered lipid concentrations associated with disease symptom severity and anti-satellite glial cell IgG antibodies. J Pain 2025, 29, 105331, doi:10.1016/j.jpain.2025.105331.
- Krock, E.; Morado-Urbina, C.E.; Menezes, J.; Hunt, M.A.; Sandstrom, A.; Kadetoff, D.; Tour, J.; Verma, V.; Kultima, K.; Haglund, L.; et al. Fibromyalgia patients with elevated levels of anti-satellite glia cell immunoglobulin G antibodies present with more severe symptoms. Pain 2023, 164, 1828-1840, doi:10.1097/j.pain.0000000000002881.

Reviewer 2 Report
Comments and Suggestions for Authors
The paper by Ueda esta al. addresses a pertinent topic in pain research, focusing on LPA₁/₃ signaling and its role in FM and NeuP. To enhance its effectiveness, consider clarifying the research gap, explicitly stating your objectives, and improving the logical flow between concepts. The paper is technically sound and of interest to a broad audience.
It would be useful of you could state how your study advances current understanding of LPA₁/₃ signaling in fibromyalgia (FM) and neuropathic pain (NeuP). A more detailed explanation of the observed sexual dimorphism in hyperalgesia, specifying the differences in LPA₁/₃ signaling pathways between male and female mice would be helpful.
Explore and discuss possible biological mechanisms that could explain the differential effects of LPA₁/₃ signaling in male and female mice, as well as the role of splenocytes in mediating hyperalgesia.
How would the results obtained contribute to the translational potential of the research? This conept in the discussion could increase the impact and relevance of the paper.
Comments on the Quality of English Language
Perhaps this is the weakest part of the paper.
I will give a few examples to be improved:
In the abstract:
1.- Original: "Since the first report that LPA1 signaling plays key roles in the initiation of nerve injury-induced neuropathic pain (NeuP), many subsequent studies have revealed that LPA1 and LPA3 signaling impact NeuP and fibromyalgia (FM) models, as revealed using knock-out mice and LPA1/3 antagonists."
Comment: The sentence is lengthy and contains redundancy ("revealed" is used twice). Consider rephrasing for clarity and conciseness.
Suggestion: "Since the initial report indicating that LPA1 signaling plays a key role in initiating nerve injury-induced neuropathic pain (NeuP), subsequent studies using knockout mice and LPA1/3 antagonists have demonstrated that LPA1 and LPA3 signaling impact NeuP and fibromyalgia (FM) models."
2.- Original: "...however intracerebroventricular treatments with Ki16425 abolished hyperalgesia in male but not female mice following."
Comment: The use of "however" requires a preceding semicolon or period. The phrase "following" is a dangling modifier and creates a sentence fragment.
Suggestion: "...; however, intracerebroventricular treatments with Ki16425 abolished hyperalgesia in male but not in female mice."
3.- Original: "Thus, these studies identify male-specific LPA1/3-mediated mechanisms in the brain underlying IPGP as well as a distinct LPA-LPA1/3-mediated peripheral immune mechanisms."
Comment: The phrase lacks parallel structure, mixing singular and plural forms.
Suggestion: "Thus, these studies identify male-specific LPA1/3-mediated mechanisms in the brain underlying IPGP, as well as distinct LPA-LPA1/3-mediated peripheral immune mechanisms."
An an example in the Discussion section:
3.- Original: "Further studies revealed that peripheral macrophage is not involved, while splenocytes derived from female model mice caused hyperalgesia in naïve mice."
Comment: Once again the phrase lacks parallel structure, mixing singular and plural forms.
Suggestion: "Further studies revealed that peripheral macrophage are not involved, while splenocytes derived from female model mice caused hyperalgesia in naïve mice."
Author Response
#Reviewer 2
Thank you very much for your kind suggestions in English explanation. As the reviewer suggested we corrected the sentences.
Comments and Suggestions for Authors
- The paper by Ueda et al. addresses a pertinent topic in pain research, focusing on LPA₁/₃ signaling and its role in FM and NeuP. To enhance its effectiveness, consider clarifying the research gap, explicitly stating your objectives, and improving the logical flow between concepts. The paper is technically sound and of interest to a broad audience.
- It would be useful of you could state how your study advances current understanding of LPA₁/₃ signaling in fibromyalgia (FM) and neuropathic pain (NeuP). A more detailed explanation of the observed sexual dimorphism in hyperalgesia, specifying the differences in LPA₁/₃ signaling pathways between male and female mice would be helpful.
>>>>According to the comments, we revised them at the last two paragraphs in the Introduction.
- Explore and discuss possible biological mechanisms that could explain the differential effects of LPA₁/₃ signaling in male and female mice, as well as the role of splenocytes in mediating hyperalgesia. How would the results obtained contribute to the translational potential of the research? This concept in the discussion could increase the impact and relevance of the paper.
>>As in the last parts of Discussion and Conclusions, we added the sentences about the future studies to clarify the molecular basis of mechanisms, which could contribute to the translational potential of the research, such as diagnosis and treatments by targeting involved molecules. (lines 364-376, 383-385)
Comments on the Quality of English Language. Perhaps this is the weakest part of the paper. I will give a few examples to be improved:
In the abstract:
- Original: "Since the first report that LPA1 signaling plays key roles in the initiation of nerve injury-induced neuropathic pain (NeuP), many subsequent studies have revealed that LPA1 and LPA3 signaling impact NeuP and fibromyalgia (FM) models, as revealed using knock-out mice and LPA1/3 antagonists." Comment: The sentence is lengthy and contains redundancy ("revealed" is used twice). Consider rephrasing for clarity and conciseness. Suggestion: "Since the initial report indicating that LPA1 signaling plays a key role in initiating nerve injury-induced neuropathic pain (NeuP), subsequent studies using knockout mice and LPA1/3 antagonists have demonstrated that LPA1 and LPA3 signaling impact NeuP and fibromyalgia (FM) models." (line 17-20)
- Original: "...however intracerebroventricular treatments with Ki16425 abolished hyperalgesia in male but not female mice following."Comment: The use of "however" requires a preceding semicolon or period. The phrase "following" is a dangling modifier and creates a sentence fragment. Suggestion: "...; however, intracerebroventricular treatments with Ki16425 abolished hyperalgesia in male but not in female mice." (line 25)
- Original: "Thus, these studies identify male-specific LPA1/3-mediated mechanisms in the brain underlying IPGP as well as a distinct LPA-LPA1/3-mediated peripheral immune mechanisms."Comment: The phrase lacks parallel structure, mixing singular and plural forms. Suggestion: "Thus, these studies identify male-specific LPA1/3-mediated mechanisms in the brain underlying IPGP, as well as distinct LPA-LPA1/3-mediated peripheral immune mechanisms." (line 30-32)
An example in the Discussion section:
- Original: "Further studies revealed that peripheral macrophage is not involved, while splenocytes derived from female model mice caused hyperalgesia in naïve mice."Comment: Once again the phrase lacks parallel structure, mixing singular and plural forms. Suggestion: "Further studies revealed that peripheral macrophage are not involved, while splenocytes derived from female model mice caused hyperalgesia in naïve mice."(line 379-318)
>>Thank you very much for kind suggestions. I revised the sentences according to these 4 suggestions.

Reviewer 3 Report
Comments and Suggestions for Authors
The manuscript presents numerous findings and is difficult to read. I have several questions.
Why is there a difference in threshold between day 0 and day 1 in Fig 1F, 1G and 1H ?
Why was ICS model used in Fig 1 for male mice and was not used in Fig 2 for female mice ?
In Fig 2BCD for day 0 why are all four thresholds the same ? The threshold should be lower for Veh and LPA antagonist.
Why were only LPA-3 KO mice used in Fig 1 whereas LPA-1 KO and LPA-3 KO mice were used in Fig 2 ? Experiments should be the same for male and female mice.
Why are Figure 1 and 2 differently organized ? The only difference should be that in Fig 1 experiments were performed on male mice whereas in Fig 2 the experiments were performed on female mice as shown in Fig 3 (the only difference between Fig 3 BCD and Fig 3 EFG is that experiments were performed on male and female mice, respectively).
In Fig 3 and 4 BCD horizontal axes are not labelled.
In Fig 1 and 2 LPA antagonist was administered for 19 days whereas in Fig 3 and 4 the antagonist was administered for 12 days. Why was it so ?
Why are results for days 0 and 1 post IPS shown in Fig 1 and 2 and not shown in Fig 3?
Why is there a difference in threshold between day 0 and day 1 in Fig 4 ?
In line 281-282 it is written that splenocytes were administered (i.v.) into naïve mice whereas Fig 6A shows that splenocytes were administered infraorbitally. Please explain this discrepancy.
In line 321-322 why were macrophages involved in abolishing the hyperalgesia and hypersensitivity in the pSNL model ?
Please clearly state in the discussion section why splenocytes derived from female IPGP model mice induce pain hypersensitivity in naïve mice.
Please clearly state in the discussion section why intracerebroventricular treatments with Ki16425 abolished hyperalgesia in male but not in female mice.
Line 26 – following what ?
In line 133 it should be minimum and not maximum I think
Line 193 – there are no Figures I and J
Line 336 remains whether is not correct English
Comments on the Quality of English LanguageThe manuscript should be edited by a naitive speaker.
Author Response
#Reviewer 3
Comments and Suggestions for Authors. The manuscript presents numerous findings and is difficult to read. I have several questions.
- Why is there a difference in threshold between day 0 and day 1 in Fig 1F, 1G and 1H
>>As the value at day 0 means the threshold at the time of before IPS treatments (Pre), and day 1 means the threshold at P1, we revised the Figures.
- Why was ICS model used in Fig 1 for male mice and was not used in Fig 2 for female mice ?
>>As previously reported, ICS and IPS model share equivalent pathophysiological and pharmacotherapeutic features, which are observed in clinic of FM patients. In the previous study [1], the IPS- or ICS-induced hyperalgesia was abolished in LPA1-KO mice. The purpose of the experiments using ICS model in the present study is just to show the involvement of LPA3 as well. However, the main purpose of the present study is about the LPA1 and LPA3 roles in the IPS-model.
- In Fig 2BCD for day 0 why are all four thresholds the same ? The threshold should be lower for Veh and LPA antagonist.
>>As mentioned above, the value at day 0 means the threshold at the time of before IPS treatments (Pre).
- Why were only LPA-3 KO mice used in Fig 1 whereas LPA-1 KO and LPA-3 KO mice were used in Fig 2 ? Experiments should be the same for male and female mice.
>>As mentioned above, we have previously reported that the IPS- or ICS-induced hyperalgesia was abolished in male LPA1-KO mice.
- Why are Figure 1 and 2 differently organized? The only difference should be that in Fig 1 experiments were performed on male mice whereas in Fig 2 the experiments were performed on female mice as shown in Fig 3 (the only difference between Fig 3 BCD and Fig 3 EFG is that experiments were performed on male and female mice, respectively).
>>As the availability of LPA1 and LPA3-KO mice is limited, we did best efforts to demonstrate minimal essential findings by these figures.
- In Fig 3 and 4 BCD horizontal axes are not labelled.
>>We added the horizontal axes.
- In Fig 1 and 2 LPA antagonist was administered for 19 days whereas in Fig 3 and 4 the antagonist was administered for 12 days. Why was it so ?
>>Since the results were already clear on the 12th, no animal testing was conducted for a long time until the 19th.
- Why are results for days 0 and 1 post IPS shown in Fig 1 and 2 and not shown in Fig 3?
>>As the condition at pre control and P1 in Fig. 3 experiments was the same as that in Fig. 1 and 2, and we focused on the sex-difference and just described the data at P5 and P12.
- Why is there a difference in threshold between day 0 and day 1 in Fig 4 ?
>>As mentioned above, the value at day 0 means the threshold at the time of before IPS treatments (Pre).
- In line 281-282 it is written that splenocytes were administered (i.v.) into naïve mice whereas Fig 6A shows that splenocytes were administered infraorbitally. Please explain this discrepancy.
>>As the reviewer pointed out, we changed the description of i.v. into the retro-orbital injection in the text and figure (lines 173 and 359) .
- In line 321-322 why were macrophages involved in abolishing the hyperalgesia and hypersensitivity in the pSNL model ?
>>In the pSNL model, macrophages will be accumulated to the damaged sensory neurons and cause some mechanisms to cause or maintain the hyperalgesia. On the other hand, as IPS model is primarily associated to the brain mechanisms, we did not know whether macrophage are involved in the IPGP mechanisms before this study.
- Please clearly state in the discussion section why splenocytes derived from female IPGP model mice induce pain hypersensitivity in naïve mice.
>>The mechanisms the hyperalgesia by splenocytes remain determined at present. But we added the description about some related possibilities in the revised manuscript, as follows.
However, it remains whether the LPA1/3-signaling in splenocytes is responsible for such mechanisms. For this discussion, it is necessary to perform several key experiments whether LPA1/3 or biosynthesis enzyme autotaxin are upregulated in splenic T-cells or B-cells in female mice treated with IPS, since there are reports that plasma cytokine levels, possibly from T-cells in FM patients seem to be associated to the FM development [2-4], and pain-related IgGs may be derived plasma cells differentiated from activated B cells [5]. Recent study revealed that IgGs from FM patients are accumulated in satellite glial cells, possibly leading to a stimulation of DRG neurons [6-8]. (line 364-376)
- Please clearly state in the discussion section why intracerebroventricular treatments with Ki16425 abolished hyperalgesia in male but not in female mice
>>In the revised manuscript (the first paragraph of the revised Discussion), we stated the sex difference in the LPA receptor-mediated brain mechanisms for hyperalgesia.
- Line 26 – following what ?
>>We deleted it.
- In line 133 it should be minimum and not maximum I think
>>We wrongly described the expression ‘maximum’. We used the average of mechanical paw withdrawal thresholds from three trials of paw pressure.
- Line 193 – there are no Figures I and J
>>We revised it. (lines 193, 198, 203, 204, and 207)
- Line 336 remains whether is not correct English
>>We revised it.
- Comments on the Quality of English Language. The manuscript should be edited by a native speaker.
>>This manuscript is revised by the British native and pain scientist (Dr. Sant-Cassia, Myles).
References
- Ueda, H.; Neyama, H. LPA1 receptor involvement in fibromyalgia-like pain induced by intermittent psychological stress, empathy. Neurobiol Pain 2017, 1, 16-25, doi:10.1016/j.ynpai.2017.04.002.
- Ellergezen, P.; Alp, A.; Cavun, S.; Celebi, M.; Macunluoglu, A.C. Pregabalin inhibits proinflammatory cytokine release in patients with fibromyalgia syndrome. Arch Rheumatol 2023, 38, 307-314, doi:10.46497/ArchRheumatol.2023.9517.
- Gerdle, B.; Dahlqvist Leinhard, O.; Lund, E.; Lundberg, P.; Forsgren, M.F.; Ghafouri, B. Pain and the biochemistry of fibromyalgia: patterns of peripheral cytokines and chemokines contribute to the differentiation between fibromyalgia and controls and are associated with pain, fat infiltration and content. Front Pain Res (Lausanne) 2024, 5, 1288024, doi:10.3389/fpain.2024.1288024.
- Peck, M.M.; Maram, R.; Mohamed, A.; Ochoa Crespo, D.; Kaur, G.; Ashraf, I.; Malik, B.H. The Influence of Pro-inflammatory Cytokines and Genetic Variants in the Development of Fibromyalgia: A Traditional Review. Cureus 2020, 12, e10276, doi:10.7759/cureus.10276.
- Goebel, A.; Krock, E.; Gentry, C.; Israel, M.R.; Jurczak, A.; Urbina, C.M.; Sandor, K.; Vastani, N.; Maurer, M.; Cuhadar, U.; et al. Passive transfer of fibromyalgia symptoms from patients to mice. J Clin Invest 2021, 131, doi:10.1172/JCI144201.
- Fanton, S.; Menezes, J.; Krock, E.; Sandstrom, A.; Tour, J.; Sandor, K.; Jurczak, A.; Hunt, M.; Baharpoor, A.; Kadetoff, D.; et al. Anti-satellite glia cell IgG antibodies in fibromyalgia patients are related to symptom severity and to metabolite concentrations in thalamus and rostral anterior cingulate cortex. Brain Behav Immun 2023, 114, 371-382, doi:10.1016/j.bbi.2023.09.003.
- Jakobsson, J.E.; Menezes, J.; Krock, E.; Hunt, M.A.; Carlsson, H.; Vaivade, A.; Emami Khoonsari, P.; Agalave, N.M.; Sandstrom, A.; Kadetoff, D.; et al. Fibromyalgia patients have altered lipid concentrations associated with disease symptom severity and anti-satellite glial cell IgG antibodies. J Pain 2025, 29, 105331, doi:10.1016/j.jpain.2025.105331.
- Krock, E.; Morado-Urbina, C.E.; Menezes, J.; Hunt, M.A.; Sandstrom, A.; Kadetoff, D.; Tour, J.; Verma, V.; Kultima, K.; Haglund, L.; et al. Fibromyalgia patients with elevated levels of anti-satellite glia cell immunoglobulin G antibodies present with more severe symptoms. Pain 2023, 164, 1828-1840, doi:10.1097/j.pain.0000000000002881.

Reviewer 4 Report
Comments and Suggestions for Authors
The manuscript presents a series of animal experiments to demonstrate the involvement of LPA1/2 receptors in the development of abnormal pain after stress stimuli. Sex difference is also confirmed. The experiments and results are interesting and soundly presented. My major concern related to the scope of the journal. The study presented would better fit to a pharmacology or neuroscience journal than a cell biology centered one.
Based on the quality of the manuscript I can support its acceptance for publication after some minor improvement. The Discussion section is a repetition of the results rather than their explanation and interpretation. The authors refer mainly to their own previous papers. More comparison to literature data and placing the results in the context of collective knowledge are recommended.
Minor language improvement is also necessary before acceptance for publication.
Author Response
#Reviewer 4
Comments and Suggestions for Authors. The manuscript presents a series of animal experiments to demonstrate the involvement of LPA1/2 receptors in the development of abnormal pain after stress stimuli. Sex difference is also confirmed. The experiments and results are interesting and soundly presented. My major concern related to the scope of the journal. The study presented would better fit to a pharmacology or neuroscience journal than a cell biology centered one.
- Based on the quality of the manuscript I can support its acceptance for publication after some minor improvement. The Discussion section is a repetition of the results rather than their explanation and interpretation.
>>We revised to add more descriptions about the explanation and interpretation of results, according to the comments by other reviewers. (line 364-376)
- The authors refer mainly to their own previous papers. More comparison to literature data and placing the results in the context of collective knowledge are recommended.
>>In the revised manuscript, we add the description and citation of literatures by other investigators.
- Minor language improvement is also necessary before acceptance for publication.
>> We asked British pain researcher for English editing.

Round 2
Reviewer 1 Report
Comments and Suggestions for Authors
I am mostly satisfied with the additions to the manuscript.
Reviewer 3 Report
Comments and Suggestions for Authors
I accept this manuscript for publication in Cells.
Comments on the Quality of English LanguageThe manuscript should be edited by a naitive speaker.